# Relationship between vitreous interleukin-6 levels and vitreous particles findings on widefield optical coherence tomography in posterior uveitis

**Mami Tomita, Mizuki Tagami** 🔗 **\*, Norihiko Misawa, Atsushi Sakai, Yusuke Haruna, Shigeru Honda** 🔗

Department of Ophthalmology and Visual Sciences, Graduate School of Medicine, Osaka Metropolitan University, Osaka, Japan

\* mizuki1979feb@yahoo.co.jp

## Abstract

### Purpose

To investigate relationship between vitreous interleukin-6 levels and vitreous particles findings on widefield optical coherence tomography in posterior uveitis.

### Methods

This retrospective study examined vitreous inflammatory cells (hyperreflective particles) of posterior uveitis on widefield optical coherence tomography (WOCT). We examined the number of hyperreflective particles (possibility of vitreous inflammatory cells) observed on WOCT and the correlations with interleukin-6 (IL-6) levels. The relationship between vitreous IL-6 levels and image findings from WOCT from 37 eyes (34 patients) with posterior uveitis were analyzed. Mean patient age was 63.4±15.7 years. (Mean± standard deviation) IL-6 concentration in vitreous humor was 79.9±7380.9 pg/mL Uveitis was infectious in 9 cases and non-infectious in 28 cases with multiplex polymerase chain reaction system. We measured the number and size of vitreous cells in the posterior vitreous, defined as the space between the upper vitreous and the internal limiting membrane on WOCT at the macular, upper, and lower regions. Image analysis software was also used for cell counting.

### Results

A strong correlation was seen between human and software counts. Pearson's correlation coefficient (PCC) was performed to compare categorial variables (on macular +0.866; upper cavity +0.713; lower cavity +0.568; total vitreous cavity +0.834; *P<0.001 each*). IL-6 levels correlated with both vitreous cell counts and cell counts observed on macular WOCT (human-counted group +0.339, *P = 0.04*; software-counted group +0.349, *P = 0.03*). Infectious uveitis showed higher IL-6 levels (*P = 0.016*) and high cell counts compared with non-infectious uveitis (*P = 0.04*).

**Data Availability Statement:** All data generated or analyzed during this study are included in this

article. Further enquiries can be directed to the corresponding author.

**Funding:** Funding: This work was supported by JSPS KAKENHI Grant Numbers 23K09013, the Charitable Trust Fund for Ophthalmic Research in Commemoration of Santen Pharmaceutical's Founder, and 2023 Osaka Community Foundation (Mizuki.Tagami.). The funders had no role in study design, data collection and analysis, decision to publish, or preparation of the manuscript.

**Competing interests:** The authors have declared that no competing interests exist.

## Conclusions

Vitreous number of hyperreflective particles (cells) findings on WOCTcorrelated well with human and software cell counts. Vitreous cells findings on WOCT also correlated with IL-6 concentrations on macular.

## Introduction

Uveitis refers to inflammation of the uvea, which comprises the iris, ciliary body and choroid. Posterior uveitis often results in vitreous opacity, leading to a loss of visual acuity. Uveitis can be infectious or non-infectious, with each requiring different approaches to therapy. Non-infectious uveitis often has a relationship with autoimmune or infectious diseases [1]. The most frequent cause of uveitis in Japan is sarcoidosis, accounting for 10.6% of cases, followed by Vogt–Koyanagi–Harada disease (8.1%), herpetic iritis (6.5%), and acute anterior uveitis (5.5%) [2], Sonoda et al. reported on 5328 patients with new-onset uveitis in 2016, finding that unclassified intraocular inflammation uveitis represented over one-third of cases, thus diagnosis and treatment of uveitis can thus be difficult. The sorting of uveitis based on the grade of inflammation may prove useful in determining therapeutic approaches.

Optical coherence tomography (OCT) is an imaging modality that possibility of enables evaluation of the inflammatory degree [3,4]. In addition, Widefield OCT (WOCT) also allows imaging of the peripheral retina comparing with normal OCT, critical inflammatory diagnosis of uveitis using non-invasive methods [5,6].

Interleukin (IL-6) is one of the cytokines involved in inflammatory reactions and immune responses [7–10]. Many recent reports have examined the relationship between uveitis and IL-6 level and several functions of IL-6 have been identified, allowing the development of biomedicines against non-infectious uveitis [11–16]. Several reports have already shown that IL-6 levels are increased in the aqueous humor in patients with uveitis [17–19].

Here we report changes in vitreous opacity in posterior uveitis as observed using WOCT imaging, IL-6 levels and their relationship. In addition to these investigations, although the number of cases is small, flow cytometry results were described in our manuscript for understanding characters of vitreous cells.

This study investigated properties and intraocular distribution of hyperreflective particles (possibility of vitreous inflammatory cells) on WOCT and the correlations with IL-6 levels and cell characters in uveitis patients.

## Materials and methods

### Selection of cases and collation of clinicopathological data

This retrospective review enrolled patients treated in the Ophthalmology Department at Osaka Metropolitan University between July 2018 and November 2022.

Study inclusion criteria included [1] the presence of posterior vitreous inflammation [2], unknown cause of posterior uveitis [3], Cases in which preoperative WOCT and Wide field Optical Coherence Tomography angiography (WOCTA) of En-face could be obtained. Study exclusion criteria included [1] Patients with active systemic inflammation [2], Cases couldn't be observed vitreous cavity due to vitreous opacity in preoperative WOCT [3], The cases couldn't take En-face images of WOCTA or Optos (Optos® 200Tx, Optos®, Dunfermline,

U.K.). We evaluated 37 eyes from 34 patients with posterior uveitis. These patients were investigated as cases of non-classified uveitis, with vitrectomy performed for diagnosis.

Data from patients were retrospectively collected. This study was performed in accordance with the tenets of the Declaration of Helsinki. All study protocols were approved by the institutional review board at Osaka Metropolitan University conducted with approval from the institutional review board (Osaka Metropolitan University-2023-043) prior to commencement. Written informed consent had been obtained from all patients enrolled in the study for the storage of patient information in the hospital database and its use in research.

All eyes underwent ophthalmologic examination, which included measurement of best-corrected visual acuity (BCVA) using a Landolt C acuity chart at 5 m, slit lamp bio-microscopy, OCT (spectral-domain OCT, Spectralis; Heidelberg Engineering, Heidelberg, Germany), and WOCT (Xephilio OCT-S1; Canon, Tokyo, Japan). BCVA was converted to logMAR BCVA.

All cases underwent 27-gauge diagnostic vitrectomy surgery and levels of IL-6 were measured.

IL-6 levels were measured in vitreous samples by enzyme-linked immunosorbent assay (ELISA) using kits for human IL-6 (LUMIPULSE G1200, FUJIREBIO, Tokyo, Japan). All vitreous samples underwent multiplex polymerase chain reaction (PCR) analysis for the diagnosis of bacterial or viral uveitis. The PCR test is a strip PCR test that can identify 24 different types of pathogenic microorganisms: herpes simplex virus (HSV) 1, HSV2, varicella-zoster virus (VZV), Epstein-Barr virus (EBV), cytomegalovirus (CMV), human herpes virus (HHV) 6, HHV7, HHV8, adenovirus, Mycobacterium tuberculosis, Treponema pallidum, human T-cell lymphotropic virus (HTLV)-1/2, Toxoplasma (T. gondii), Toxocara, Chlamydia trachomatis (C. trachomatis), Propionibacterium acnes (P. acnes), Aspergillus, Fusarium, bacterial 16S ribosomal RNA (rRNA), Candida species (Candida sp.), C. glabrata, C. krusei, fungal 28S rRNA, and Acanthamoeba [20].

When bacteria or virus was detected in vitreous samples, infectious uveitis was diagnosed. If no findings or results of morbid significance were obtained, the case was classified as non-infectious uveitis.

We observed the vitreous area on WOCT before vitrectomy and defined hyperreflective particles as cells [3]. Also, En-face WOCTA was taken in all cases. Macular was defined horizontal B-scan through the foveal central subfield [3,21]. Furthermore, we examined the choroidal structure using WOCTA or Optos 200Tx (Optos PLC, Dunfermline, Scotland) [22,23]. We identified the ampullae of the vortex vein inflow by imaging in En-face choroidal mode of OCTA or En-face mode of Optos. The B-scan around equatorial region was defined as a straight line connecting temporal and nasal of the ampullae vortex vein inflow [24]. Upper vitreous cavity area on WOCT was defined as B-scan with horizontal lines drawn at the inflow point of the vortex vein volume on the superotemporal and superonasal sides. For the lower vitreous cavity area of WOCT, like the upper one, we selected a B-scan around the area connecting inferotemporal and inferonasal sides with a horizontal line. If the temporal and nasal vortex venous ampullaes could not be connected horizontally, a horizontal line passing through the nasal vortex vein ampulla was used.

Cells observed on WOCT were counted for B-scan images through the macular (Fig 1A), upper vitreous cavity area around the equatorial region (Fig 1B) and lower vitreous cavity area (Fig 1C). Vitreous cells were counted in the vitreous space using WOCT by humans. Furthermore, the utility of ImageJ software was examined. We manually selected the vitreous area (Fig 1D–1F), if areas obviously depicting vitreous were misidentified as cells, the vitreous membrane was manually selected to be avoided. (Fig 1E and 1F). Then counted the number of cells present in that area using ImageJ software application (Fig 1G–1I). Measurements were made with reference to previous reports [25].

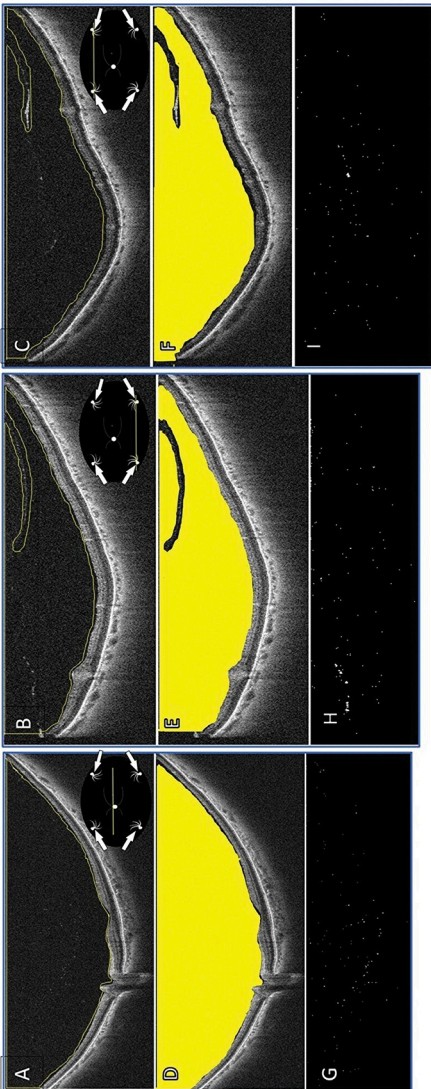

**Fig 1. Measurement of the number of hyperreflective particles on WOCT.** (A) The horizontal WOCT through the macular. (B) The horizontal WOCT through upper vitreous cavity area around the equatorial region. (C) The horizontal WOCT through lower vitreous cavity area around the equatorial region. (D) The area where measurements were actually taken in the macular part on WOCT. (E) The area where measurements were actually taken in the upper part on WOCT. (F) The area where measurements were actually taken in the lower part on WOCT. (G) The vitreous cells analyzed by software on WOCT through the macular. (H) The vitreous cells analyzed by software on WOCT through the upper. (I) The vitreous cells analyzed by software on WOCT through the lower. White arrow: Vortex vein inflow. We selected the vitreous area manually using to selection tools in ImageJ (A-C). We selected and analyzed "vitreous area" using ImageJ software (D-F). ImageJ software analyzed hyperreflective particles (G-I).

WOCT images were taken horizontally at a maximum of 23 mm and at a maximum depth of 5.3 mm; the average measured vitreous cavity area in 37 cases was 22.5 x 3.0 mm through the macular, 16.8 x 4.4 mm around upper around equatorial area, 15.9 x 3.8 mm around lower.

We compared the number of cells observed on WOCT with IL-6 levels and examined differences between infectious or non-infectious uveitis. All data were collected in an Excel spreadsheet (Microsoft, Redmond, WA, USA).

## Imaging evaluations

In the vitreous area, hyperreflective particles were counted one by one.

Images in the lower parts of three cases were unclear in the delineation and were excluded from analysis.

In accordance with previous reports, "posterior vitreous cavity area" was measured as the space between the upper edge of the vitreous body and the inner boundary membrane, after manual drawing using the polygon selection tool in ImageJ software [3,25]. In this study, the presence or absence of posterior vitreous detachment was not considered. Data obtained from an independent scorer who had been blinded to cases were used for analysis. All WOCT images were imported into ImageJ software version 1.5 (National Institutes of Health, Bethesda, MD, USA). All assessments were applied to each image; image contrast was not adjusted before or after exporting the image from the OCT system.

Our WOCT device that could obtain up to 23 mm of widefield B-scan images in a single acquisition. The device had a depth of 5.3 mm. The transverse resolution was 20 μm.

The system has a digital axial resolution of 1.6 μm, which can be combined with the ability to average multiple scans (up to 200) to further improve image quality.

Two methods of counting were used: counting by human (Group 1); and counting by software (Group 2). We used ImageJ software for counting. ImageJ is a Java-based image-processing program developed by the National Institutes of Health and the Optical Computational Metrology Institute. We manually selected the vitreous area using the selection tools in ImageJ. Binarization is a type of image processing that divides the intensity of an image into binary values, such as 0 or 1 [26]. On the WOCT images, cells were binarised and automatically counted by the ImageJ software [27]. The image area to be binarized needs to be manually selected. The correlation between cell counts and IL-6 levels in the images from each case was analyzed in each group.

## Correlation with IL-6 value

We compared logMAR BCVA between pre-vitrectomy and 3 months postoperatively, and examined relationships between BCVA and the number of cells.

IL-6 levels were separated into below 1000 and above 1001 pg/mL, then we investigated whether cell counts observed from WOCT were high.

We also investigated logMAR BCVA preoperatively and at 3 months postoperatively. The group showing improved vision was defined as those achieving two or more levels of improvement in BCVA. The group with no changes in visual acuity was defined as the unchanged group. Those patients in whom visual acuity worsened by two or more levels between preoperatively and 3 months postoperatively were defined as the worsening group in each group.

## Flow cytometry

In flow cytometry, gating is the process of selecting a subset of cells from a larger population for further analysis. Gating was performed by setting up a series of gates or regions on a plot of the data generated by the flow cytometer.

They separate cell fractions using forward scatter (FSC) and side scatter (SSC). High score of FSC means that cells in the samples is large. Large cells were identified as gate(A).

We analyzed using the CD45-SSC gating method. The CD45-SCC gating method uses the fact that immature, proliferative haematopoietic cells express low levels the CD45 antigen. This allows selective analysis of tumor cells by excluding mature normal cells [28].

CD45 gating were enable to analysis CD45 diminish area as Gate1.

**Table 1. Baseline characteristics of patients.**

| Participant Baseline Characteristics | total | infectious | Non-infectious |
|---|---|---|---|
| Eye,n (%) | 37 | 9 (24.3) | 28 (75.7) |
| Patients,n (%) | 34 | 9 (26.5) | 25 (73.5) |
| Males,n (%) | 18 | 8 (44.4) | 10 (55.6) |
| Females,n (%) | 19 | 1 (5) | 18 (95) |
| Age,years (Mean±SD) | 67.5±15.7 | 67±18.8 | 67.5±14.2 |
| Baseline logMAR BCVA (Mean±SD) | 0.22±0.6 | 0.15±0.8 | 0.35±0.5 |
| IL-6 levels, pg/mL (Mean±SD) | 79.9±7380.9 | 1040±12990.6 | 73.5±990.5 |

Abbreviation: BCVA: Best-corrected visual acuity, SD: Standard deviation, IL-6: Interleukin-6.

The number of cells observed in Gate 1 was calculated and compared with the number of cells observed on WOCT. This analysis included subjects showing positivity for CD3, CD8, CD19, CD20 and CD56. The composition of lymphocytes are shown. CD19 and CD20 represent the B-cell family and were investigated together. Seven cases in enough vitreous fluid were analyzed by flow cytometry analysis for detecting vitreous cells characters.

## Statistical analyses

Clinical and histopathological characteristics are summarized using descriptive statistics. Correlations between immunohistochemical, demographic, and clinicopathological factor data were assessed using the t-test and chi-squared test. Statistical analyses were performed using SPSS Statistics software (version 22; IBM Japan, Tokyo, Japan). Values of $P<0.05$ were considered significant. IL-6 levels and cell counts observed on OCT were analyzed using the Mann-Whitney U-test.

## Results

In total, 37 eyes from 37 patients were included in the study "Table 1".

Table 1 shows patient characteristics. PCR revealed the cause of infection as acute retinal necrosis in 3 cases, cytomegalovirus in 1 case, bacterial (16-strip PCR) in 1 case, and human T-cell leukemia virus type 1 in 1 case. Almost all cases of non-infectious uveitis were idiopathic (n = 22), with 5 cases caused by sarcoidosis and 1 case caused by macular pit syndrome.

On the WOCT images, the more cells there were in Group 1, the more cells were observed in Group 2 as well. (Table 2, Fig 2A–2D).

**Table 2. Numbers of cells in Group 1 and Group 2.**

| | Group 1 (n) | Group 2 (n) | PCC | P-value |
|---|---|---|---|---|
| Macular | 27.7±45.0 | 95.5±76.3 | +0.866 | <0.001* |
| Upper | 8.5±22.9 | 61.5±53.8 | +0.713 | <0.001* |
| Lower (n = 34) | 8.7±17.0 | 56.7±37.7 | +0.568 | <0.001* |
| Total (n = 34) | 44.0±73.0 | 209.1±133.4 | +0.834 | <0.001* |

Statistical analysis were performed using the Pearson's correlation coefficient.

Group 1: Manual counting, Group 2: Software-based counting.

PCC: Pearson's correlation coefficient.

*$P<0.05$.

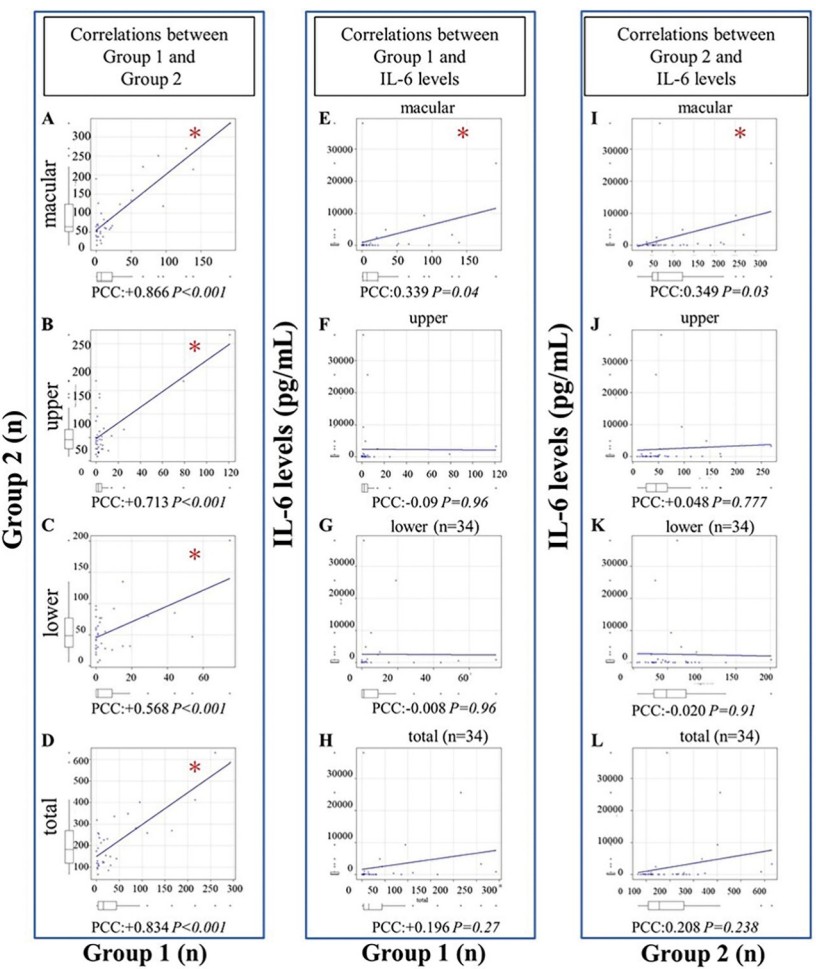

**Fig 2. Relationship between Group 1 and Group 2.** Group 1: Manual counting, Group 2: Software-based counting. (A) In macular images, correlations between Group 1 and Group 2. (B) In upper vitreous around the equator images, correlations between Group 1 and Group 2. (C) In lower vitreous around the equator images, correlations between Group 1 and Group 2. (D) In total number of cells analyzed in the macula, upper and lower, correlations between Group 1 and Group 2. (E) In macular images, correlations between Group 1 and IL-6 levels. (F) In upper vitreous around the equator images, correlations between Group 1 and IL-6 levels. (G) In lower vitreous around the equator images, correlations between Group 1 and IL-6 levels (n = 34). (H) In total number of cells analyzed in the macula, upper and lower, correlations between Group 1 and IL-6 levels (n = 34). (I) In macular images, correlations between Group 2 and IL-6 levels. (J) In upper vitreous around the equator images, correlations between Group 2 and IL-6 levels. (K) In lower vitreous around the equator images, correlations between Group 2 and IL-6 level (n = 34). (L) In total number of cells analyzed in the macula, upper and lower, correlations between Group 1 and IL-6 levels (n = 34). PCC: Pearson's correlation coefficient, IL-6: Interleukin-6. *$P<0.05$.

Cell counts were consistently higher in Group 2 than in Group 1. Each image showed a high correlation coefficient between groups.

All area showed significant findings. As more cells were observed in Group 1, more cells were also found in Group 2. We investigated the correlation between IL-6 levels and cell counts on WOCT in Group 1 (Fig 2E–2H). We found a correlation between IL-6 level and cell count in Group 1 from macular images (PCC +0.339, $P = 0.04$; Fig 2E). We got correlations between IL-6 levels and cell counts from WOCT in Group2 using "ImageJ" software (Fig 2I–2L). Likewise, in Group 2, significant differences were found only in macular images (PCC +0.349, $P = 0.03$; Fig 2I).

**Table 3. Correlation between IL-6 levels and cell counts on WOCT images.**

| | Group 1 PCC | P-value | Group 2 PCC | P-value |
|---|---|---|---|---|
| Macular | +0.339 | 0.0402* | +0.349 | 0.0343* |
| Upper | -0.009 | 0.96 | +0.048 | 0.777 |
| Lower (n = 34) | -0.008 | 0.96 | -0.020 | 0.914 |
| Total (n = 34) | +0.196 | 0.266 | +0.208 | 0.238 |

Statistical analysis were performed using the Pearson's correlation coefficient.

Group 1: Manual counting, Group2: Software-based counting.

PCC: Pearson's correlation coefficient.

*P<0.05.

The number of cells in macular images thus correlated with IL-6 levels in both groups (Table 3).

Only images at the macular showed a significant correlation (P = 0.0402; Group1, P = 0.0343; Group2).

We added a study of macular images based on this high correlation between IL-6 levels and cell count. As a high correlation was found between Groups 1 and 2, further analyses were added for only the mechanical analysis group.

In IL-6 levels, no significant relationship was seen between IL-6 levels and number of cells, but higher cell numbers tended to be observed in cases with higher IL-6 levels (*P = 0.068*; Fig 3A).

The 37 patients were divided into groups with infectious and non-infectious uveitis. In each group, we compared IL-6 levels and cell counts from macular images. The infectious uveitis group showed higher IL-6 levels than the non-infectious group (*P<0.001*; Fig 3B). This result was similar to findings from a previous report [29]. Mean IL-6 levels was 8689.6±13,778.6 pg/mL in the infectious group and 387.8±1008.7 pg/mL in the non-infectious group. We also compared cell counts between the infectious and non-infectious uveitis groups. Among infectious cases, many cells were seen in WOCT images, as previously reported (*P = -0.021*; Fig 3C) [25]. Mean cell count was 161.3±113.2 units in the infectious group and 74.4±47.6 units in the non-infectious group.

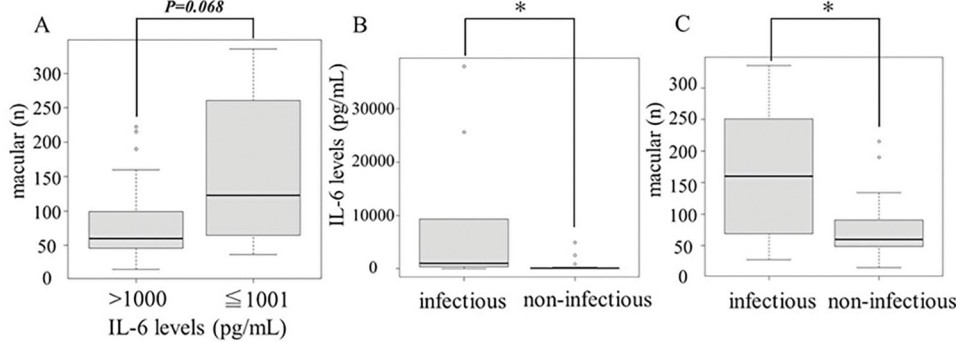

**Fig 3.** Relationship between IL-6 levels and the number of cells when the IL-6 value is bimilitarised (A). The number of vitreous particles as hyperreflective particles in infectious and non-infectious uveitis (B, C). (A) We separated two groups according to IL-6 levels below 1000 pg/mL above 1001 pg/mL. IL-6 levels tended to be higher with greater cell counts on WOCT, but the relationship was not significant using the t-test. (B) IL-6 levels were higher in infectious uveitis compared with noninfectious uveitis using the Mann-Whitney test. (C) Patients with infectious uveitis showed significantly greater cell counts than those with non-infectious uveitis using the t-test. IL-6: Interleukin-6. *P<0.05.

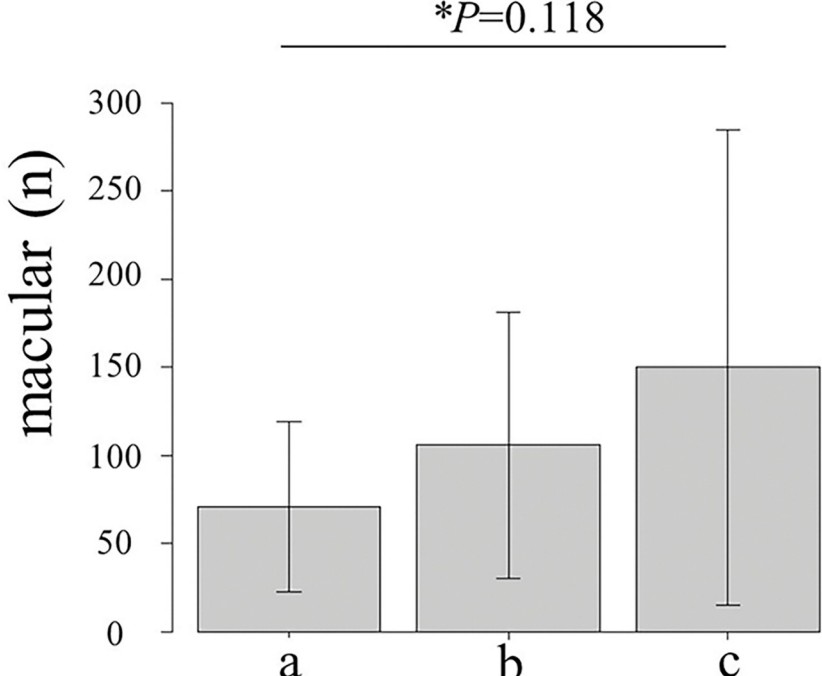

**Fig 4. Comparisons of IL-6 levels among three groups depending on improvement or deterioration of visual acuity.** a) Group in which visual acuity improved by ≥2 grades (70.8±48.4). b) Group in which visual acuity was unchanged (105.6±75.8). c) Group in which visual acuity worsened by ≥2 grades (149.8±134.9). *One-way Analysis of Variance (ANOVA).

## Correlation with IL-6 levels and changing of visual acuity

IL-6 levels were separated into below 1000 and above 1001 pg/mL, then we investigated whether cell counts observed from WOCT were high.

We also investigated logMAR BCVA preoperatively and at 3 months postoperatively. ($P = 0.018$; Fig 4).

We also considered the relationship between number of cells on WOCT images and lymphocyte count from flow cytometry (Table 4).

There was no statistically significant difference between the observed the number of cells on WOCT and each lymphocyte (Fig 5).

**Table 4. Baseline characteristics of the 7 patients investigated by flow cytometry.**

| ID | sex | Age(years) | classification | The number of cells in WOCT (macular) | CD3 (n) | CD8 (n) | CD19+20 (n) | CD56 (n) |
|----|-----|-----------|----------------|---------------------------------------|---------|---------|-------------|----------|
| 1 | F | 56 | PIOL | 38 | 48.5 | 41.2 | 13.8 | 3.6 |
| 2 | F | 76 | Sarcoidosis | 15 | 103.9 | 1.5 | 2.0 | 4.2 |
| 3 | F | 47 | Idiopathic | 123 | 44.3 | 10.0 | 6.9 | 4.2 |
| 4 | M | 44 | Idiopathic | 46 | 29.9 | 42.3 | 4.2 | 1.3 |
| 5 | F | 69 | Idiopathic | 28 | 338.1 | 18.7 | 7.5 | 3.6 |
| 6 | M | 80 | Idiopathic | 53 | 31.3 | 42.7 | 228.7 | 7.8 |
| 7 | M | 57 | Idiopathic | 51 | 16.0 | 122.9 | 122.9 | 5.7 |

Abbreviations: PIOL: Primary Intraocular lymphoma.

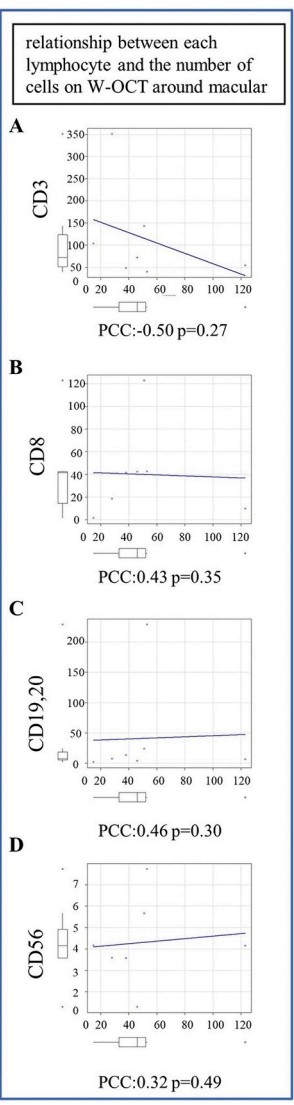

**Fig 5. The number of lymphocytes as observed by flow cytometry was compared with cell counts on WOCT.** (A) Comparison between cells and positivity for CD3. (B) Comparison between cells and positivity for CD8. (C) Comparison between cells and positivity for CD19+20. (D) Comparison between cells and positivity for CD56.

## Discussion

We performed vitrectomy in 37 cases of unclassified posterior uveitis, and analyzed IL-6 levels from vitreous samples. We also observed reflective particles on B-scan widefield OCT performed pre-vitrectomy. This study identified significant differences in the number of vitreous hyperreflective particles on WOCT images of the macular according to IL-6 levels.

Diagnostic vitrectomy is a very general and important clinical tool for diagnosing uveitis. The safety and utility of 27-gauge vitrectomy have recently been reported [30].

Focusing on molecular uveitis, IL-6 is a cytokine showing elevated production in autoimmune diseases and infections [31,32]. We have previously described some associations with IL-6 levels in posterior uveitis [29,30,33]. In addition, other reports have found elevated IL-6 in macular edema associated with uveitis, and inhibition of IL-6 may be an effective method

for treating non-infectious uveitis [34–36]. However, the mechanisms by which inflammatory cells migrate into the eye during uveitis are not yet fully understood.

IL-6 mainly produced by T cells, B cells and monocytes/macrophages in the eyes with uveitis.

Other reports have examined where IL-6 is produced, such as the retinal pigment epithelium (RPE), Muller cells or retinal endothelial cells [37–39]. IL-6 has been reported to produce vascular endothelial growth factor (VEGF) [40–43]. VEGF is a signaling protein that plays a crucial role in promoting the growth of new blood vessels.

Signal transducer and activator of transcription (STAT) proteins are a family of cytoplasmic transcription factors that play important roles in signal transduction and gene expression regulation. STAT is phosphorylated by janus kinase (JAK), STAT regulate the expression of related genes.

This pathway is known as the Janus kinase/transcriptional activation signaling pathway (JAK/STAT) signaling pathway.

VEGF is said to be associated with the JAK/STAT pathway, although the underlying mechanisms remain unclear.

IL-6 also leads to the production of intercellular adhesion molecule1 (ICAM-1) [44–46]. ICAM-1 is a cell surface glycoprotein, acting as an adhesion receptor that is well known for controlling the recruitment of leukocytes from the circulation to sites of inflammation [47].

Both VEGF and ICAM-1 increase vascular permeability, and VEGF promotes neovascularization. Increased vascular permeability may thus be involved in the findings of vitreous cell infiltration and strong inflammation.

OCT is an important tool in clinical practice in ophthalmology. The resolution achievable by OCT has been increasing recently, and rapid imaging has become possible. In the treatment of uveitis, this is very important [48,49]. Currently, OCT is often used to determine treatment strategies. WOCT is capable of imaging the retina at a wide angle, allowing evaluation of the peripheral retina. Articles reporting on the relationships between WOCT and clinical findings are increasing [5,6]. Recent studies have attempted to analyze OCT from various angles using image analysis software.

In this study, we evaluated WOCT findings using the binarization function of ImageJ. We were able to detect vitreous cells from images using binarization.

We have reported for the first time that vitreous cells in the vitreous area of WOCT images were binarized using ImageJ, and comparison with IL-6 levels yielded significant results. When human observers counted the number of highly reflective particles on macular WOCT, correlations with IL-6 levels were similar to those seen from software counting. In this study, Group 1 (manual counting) had less the number cells on WOCT than Group 2 (software counting). Also, there were large differences in the number of cells counted between Group 1 and 2. This suggests that software may be able to capture brightness that humans cannot see visually. In all WOCT images of macular, upper and lower, there was a positive correlation between the number of cells counted by humans and the number of cells analyzed by the software.

Here we reported that hyperreflective particles in the vitreous cavity at the macular correlated with vitreous IL-6 levels.

Patients with macular edema and some severe retinal diseases have been shown to have high IL-6 levels, so the present results suggest that cell counts on WOCT might be involved with the severity of pathology [32]. Although no significant results were identified in this study, WOCT may be used in the future to evaluate the prognosis of visual acuity.

This study examined only IL-6, but many other inflammatory cytokines may be related to uveitis, and we intend to focus on other cytokines and interleukins in the future.

## Limitations

All patients in this study were Asian, including one Vietnamese patient. The remaining patients were Japanese. The generalizability of the study findings to other ethnic groups needs to be examined in the future.

The study design was limited by the retrospective design and the relatively small number of subjects. Given the small sample size and the inclusion of patients from a single ethnic background, this study cannot be considered representative or to comprehensively cover all types of uveitis. Another limitation was various issues with images. The first problem is that the posterior vitreous cavity was selected for image analysis. Strictly speaking, the posterior vitreous cavity area includes the posterior vitreous precortical pocket, which is considered distinct from the vitreous area. We think that it is an important issue that the "posterior vitreous area" that is measured is not fair depending on the presence or absence of posterior vitreous detachment. A distinction between vitreous pockets and posterior vitreous areas should be considered in the future. Second, this study was performed using manual procedures. Selecting the vitreous cavity manually may introduce bias. Third, WOCT has a de-noising function, and sometimes does not capture images as is. The problem with this case series is the WOCT poor images. Images obtained by WOCT were not corrected. This is because it was thought that image correction would become arbitrary. The poor image is one of the problems, however we couldn't find good idea to solve it. Further, vitreous cells are not always captured in a single shot. As further cases are accumulated, future investigations will be able to determine whether hyperreflective particles in WOCT images correlate with the prognosis of visual function.

## Conclusions

Vitreous number of hyperreflective particles (cells) findings on WOCT correlated well with human and software cell counts. Vitreous cells findings on WOCT also correlated with IL-6 concentrations on macular.

## Supporting information

**S1 File.**
(XLSX)

## Author Contributions

**Conceptualization:** Mizuki Tagami.

**Data curation:** Mami Tomita, Mizuki Tagami.

**Formal analysis:** Mami Tomita, Mizuki Tagami, Norihiko Misawa.

**Funding acquisition:** Mizuki Tagami.

**Investigation:** Mami Tomita, Mizuki Tagami, Norihiko Misawa, Atsushi Sakai, Yusuke Haruna.

**Methodology:** Mami Tomita, Mizuki Tagami.

**Project administration:** Mizuki Tagami.

**Software:** Mami Tomita.

**Supervision:** Shigeru Honda.

**Validation:** Mizuki Tagami.

**Visualization:** Mizuki Tagami.

**Writing – original draft:** Mami Tomita, Mizuki Tagami.

**Writing – review & editing:** Mizuki Tagami, Shigeru Honda.

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
