## [Decision Letter · Decision Letter 0]

6 Nov 2023

PONE-D-23-27793Relationship between vitreous interleukin-6 levels and vitreous particles findings on widefield optical coherence tomography in posterior uveitisPLOS ONE

Dear Dr. Tagami,

Thank you for submitting your manuscript to PLOS ONE. After careful consideration, we feel that it has merit but does not fully meet PLOS ONE’s publication criteria as it currently stands. Therefore, we invite you to submit a revised version of the manuscript that addresses the points raised during the review process.

Please submit your revised manuscript by Dec 21 2023 11:59PM.  If you will need more time than this to complete your revisions, please reply to this message or contact the journal office at plosone@plos.org. Please include the following items when submitting your revised manuscript:A rebuttal letter that responds to each point raised by the academic editor and reviewer(s). You should upload this letter as a separate file labeled 'Response to Reviewers'.A marked-up copy of your manuscript that highlights changes made to the original version. You should upload this as a separate file labeled 'Revised Manuscript with Track Changes'.An unmarked version of your revised paper without tracked changes. You should upload this as a separate file labeled 'Manuscript'.

We look forward to receiving your revised manuscript.

Kind regards,

Jiro Kogo

Academic Editor

PLOS ONE

Journal Requirements:

"Funding: This work was supported by JSPS KAKENHI Grant Numbers 23K09013 , the Charitable Trust Fund for Ophthalmic Research in Commemoration of Santen Pharmaceutical’s Founder, and 2023 Osaka Community Foundation (Mizuki.Tagami.)."

6. Please expand the acronym “JSPS” (as indicated in your financial disclosure) so that it states the name of your funders in full.

7. Please upload a new copy of Figure 3 as the detail is not clear. Please follow the link for more information: " ext-link-type="uri" xlink:type="simple">https://blogs.plos.org/plos/2019/06/looking-good-tips-for-creating-your-plos-figures-graphics/"
" ext-link-type="uri" xlink:type="simple">https://blogs.plos.org/plos/2019/06/looking-good-tips-for-creating-your-plos-figures-graphics/"

Reviewers' comments:

Reviewer's Responses to Questions

**Comments to the Author**

1. Is the manuscript technically sound, and do the data support the conclusions?

Reviewer #1: Yes

Reviewer #2: Partly

2. Has the statistical analysis been performed appropriately and rigorously? 

Reviewer #1: No

Reviewer #2: Yes

3. Have the authors made all data underlying the findings in their manuscript fully available?

Reviewer #1: Yes

Reviewer #2: Yes

4. Is the manuscript presented in an intelligible fashion and written in standard English?

Reviewer #1: No

Reviewer #2: Yes

5. Review Comments to the Author

Reviewer #1: The authors demonstrated that relationship between vitreous interleukin-6 levels and vitreous particles findings on widefield optical coherence tomography in posterior uveitis. The manuscript contains many grammatical errors. In addition, the purpose of the study is ambiguous. The methods for counting vitreous cells are not clear. Major revision must be necessary.

1. Line 25-26. Purpose of this study is ambiguous. Is the purpose may be to clarify the relationship between vitreous interleukin-6 levels and vitreous particles findings on widefield optical coherence tomography in posterior uveitis?

2. Line 56. “and acute anterior uveitis (5.5%),” should be “and acute anterior uveitis (5.5%) [2].”

3. Line 58. “represented over half of case, thus Diagnosis and treatment of” should be “represented over one-third of cases, thus diagnosis and treatment of”.

4. Line 62. “Widefield OCT angiography (WOCTA)” should be “Widefield OCT (WOCT)”.

5. Line 66-67. “the relationship between uveitis and IL-6 level and several functions of IL-6 have been identified” should be “relationship between uveitis and IL-6 levels in aqueous humor and significant roles of IL-6 in non-infectious uveitis have been identified”.

6. Line 69. “increased in uveitis” should be “increased in the aqueous humor in patients with uveitis”.

7. Line99-100. Please describe the details of multiplex PCR. What kinds of pathogens did you examine? Did you examine fugus or toxoplasma DNA? It is necessary to include references related to multiplex PCR.

8. Line 104-115. I cannot understand the meaning of Figure 1A, 1B, 1C, 1D, 1E and 1F. Where is the area of “through the macula (Figure 1A)”, “upper vitreous cavity around the equatorial region (Figure 1B)” and “lower vitreous cavity (Figure 1C)”? How to determine the position of the equator in a WOCT image?

9. Line 131-133. What is the area for counting cells in vitreous cavity in a WOCT image? (for example, 13mm x 5mm)

10. Line 140-142. If you defined “posterior vitreous area” as described here, the area for counting the number of cells varies greatly depending on the height of the PVD, and this is not fair for evaluation of the severity of inflammation in the vitreous cavity.

11. Line 150-151. The methods for “counting by software” should be described. References for the methods for “counting by software” are necessary.

12. Line 167-168. In flow cytometry analyses, it is common to separate cell fractions (erythrocytes, lymphocytes, monocytes, and granulocytes) using forward scatter and side scatter at first. Which cell fraction did you gate on?

13. Line 194-202. Numbers of cells in Group 1 (manual counting) and Group 2 (software-based counting) were quite different and more numbers of cells was counted by manual counting. Why is there such a big difference between manual counting and software-based counting?

14. Line 268. “Correlation with IL-6 value” should be “Correlation with IL-6 levels and changing of visual acuity”.

15. Line 277. “Fig 4. We compared visual acuity between preoperatively and 3 months postoperatively“ should be “Fig 4. Comparisons of IL-6 levels among three groups depending on improvement or deterioration of visual acuity”.

16. Line 283-291. The finding that there is a correlation between the number of cells in WOCT and CD56-positive cells in FACS is clearly due to the fact that one case of ARN had an exceptionally high number of CD56-positive cells. ARN is infectious uveitis, and the other eight cases are non-infectious. If you want to claim that there is a correlation between the cell count in WOCT and CD56-positive cells in FACS, you should exclude one case of ARN from the analysis.

17. Line 311-313. IL-6 may be mainly produced by T cells, B cells and monocytes/macrophages in the eyes with uveitis.

18. Line 329. “OCT is an important tool in ophthalmology” should be “OCT is an important tool in clinical practice in ophthalmology”.

19. Line 332. “Papers” should be “Articles”.

20. Line 337. “We evaluated WOCT findings” should be “In this study, we evaluated WOCT findings”.

21. Line 337-340. The methods for binarization of WOCT image and the methods for counting vitreous cells should be described in Materials and Methods section.

22. Line 368. “WOCTA” should be “WOCT”.

Reviewer #2: 3.11.2023

In this manuscript, the authors described “Relationship between vitreous interleukin-6 levels and vitreous particles findings on widefield optical coherence tomography in posterior uveitis”. The study will be very useful for the literature. The report is an interesting study, but it needed some suggestions for publication.

Here are the concerns for the authors;

Generally;

1. Keywords should be ordered alphabetical.

2. Grammar should be checked.

3. Abstract structure should be the same as described in the journal.

4. The inclusion and exclusion criteria determined for the study should be clearly stated in the materials and methods.

5. The findings given in the table should not be repeated in the text in the results section.

6. The units of evaluated parameters should be given clearly. For example, pg/mL for IL-6 value.

7. The % of demographic characteristics should also be given in Table 1.

8. The spelling features of Tables 1 and 2 are not the same.

9. All p values should be written in italic.

10. The resolution of the figures is low and should be corrected.

11. Citations should be corrected according to the journal instructions.

12. The lot number of kits should be stated.

13. The brand and model numbers of the devices used in the study should be stated.

14. What does the word “patiene” in Table 1 mean?

15. Significance levels of p values should be stated below the tables.

6. PLOS authors have the option to publish the peer review history of their article (what does this mean?). If published, this will include your full peer review and any attached files.

Reviewer #1: **Yes: **Toshikatsu Kaburaki

Reviewer #2: No

---

## [Author Response · Author response to Decision Letter 0]

4 Dec 2023

Journal Requirements:

・ Please ensure that your manuscript meets PLOS ONE's style requirements, including those for file naming. The PLOS ONE style templates can be found at 

Response

Thank you for your advice. We modified sentences.

・ Note from Emily Chenette, Editor in Chief of PLOS ONE, and Iain Hrynaszkiewicz, Director of Open Research Solutions at PLOS: Did you know that depositing data in a repository is associated with up to a 25% citation advantage (https://doi.org/10.1371/journal.pone.0230416)? If you’ve not already done so, consider depositing your raw data in a repository to ensure your work is read, appreciated and cited by the largest possible audience. You’ll also earn an Accessible Data icon on your published paper if you deposit your data in any participating repository (https://plos.org/open-science/open-data/#accessible-data).

Response

Thank you for your advice. This time, we decided not to deposit data into participating repositories.

・ Thank you for stating the following financial disclosure: 

"Funding: This work was supported by JSPS KAKENHI Grant Numbers 23K09013 , the Charitable Trust Fund for Ophthalmic Research in Commemoration of Santen Pharmaceutical’s Founder, and 2023 Osaka Community Foundation (Mizuki.Tagami.)."

Response

We added the following. 

L605-606

・ Please upload a new copy of Figure 3 as the detail is not clear. Please follow the link for more information: https://blogs.plos.org/plos/2019/06/looking-good-tips-for-creating-your-plos-figures-graphics/" https://blogs.plos.org/plos/2019/06/looking-good-tips-for-creating-your-plos-figures-graphics/"

Response

Thank you for pointing out. Figure 3 was uploaded a new copy.

#Reviewer1

・ Line 25-26. Purpose of this study is ambiguous. Is the purpose may be to clarify the relationship between vitreous interleukin-6 levels and vitreous particles findings on widefield optical coherence tomography in posterior uveitis?

Response

Thank you for your advice. We modified as the following.

L18-19

Purpose: To investigate relationship between vitreous interleukin-6 levels and vitreous particles findings on widefield optical coherence tomography in posterior uveitis

・ Line 56. “and acute anterior uveitis (5.5%),” should be “and acute anterior uveitis (5.5%) [2].”

Response

Thank you for your advice. We corrected the position of the word.

・ Line 58. “represented over half of case, thus Diagnosis and treatment of” should be “represented over one-third of cases, thus diagnosis and treatment of”.

Response

Thank you for your advice. We modified the sentences.

・ Line 62. “Widefield OCT angiography (WOCTA)” should be “Widefield OCT (WOCT)”.

Response

Thank you for your comment. We corrected the word.

・ Line 66-67. “the relationship between uveitis and IL-6 level and several functions of IL-6 have been identified” should be “relationship between uveitis and IL-6 levels in aqueous humor and significant roles of IL-6 in non-infectious uveitis have been identified”

Response

Thank you for your advice. We modified the sentences.

・ Line 69. “increased in uveitis” should be “increased in the aqueous humor in patients with uveitis”.

Response

Thank you for your advice. We corrected words.

・ Line99-100. Please describe the details of multiplex PCR. What kinds of pathogens did you examine? Did you examine fugus or toxoplasma DNA? It is necessary to include references related to multiplex PCR.

Response

Thank you for your advice. We added the following.

L99-105

The PCR test is a strip PCR test that can identify 24 different types of pathogenic microorganisms: herpes simplex virus (HSV) 1, HSV2, varicella-zoster virus (VZV), Epstein-Barr virus (EBV), cytomegalovirus (CMV), human herpes virus (HHV) 6, HHV7, HHV8, adenovirus, Mycobacterium tuberculosis, Treponema pallidum, human T-cell lymphotropic virus (HTLV)-1/2, Toxoplasma (T. gondii), Toxocara, Chlamydia trachomatis (C. trachomatis), Propionibacterium acnes (P. acnes), Aspergillus, Fusarium, bacterial 16S ribosomal RNA (rRNA), Candida species (Candida sp.), C. glabrata, C. krusei, fungal 28S rRNA, and Acanthamoeba[20].

・ Line 104-115. I cannot understand the meaning of Figure 1A, 1B, 1C, 1D, 1E and 1F. Where is the area of “through the macula (Figure 1A)”, “upper vitreous cavity around the equatorial region (Figure 1B)” and “lower vitreous cavity (Figure 1C)”? How to determine the position of the equator in a WOCT image?

Response

Thank you for pointing out. We modified any definition as following.

Also, the resolution of Fig1 was corrected and some schema were added to show which area the B-scan image cavers.

L110-120

Also, En-face WOCTA was taken in all cases. Macular was defined horizontal B-scan through the foveal central subfield [3, 21]. Furthermore, we examined the choroidal structure using WOCTA or Optos 200Tx (Optos PLC, Dunfermline, Scotland) [22, 23]. We identified the ampullae of the vortex vein inflow by imaging in En-face choroidal mode of OCTA or En-face mode of Optos. The B-scan around equatorial region was defined as a straight line connecting temporal and nasal of the ampullae vortex vein inflow[24]. Upper vitreous cavity area on WOCT was defined as B-scan with horizontal lines drawn at the inflow point of the vortex vein volume on the superotemporal and superonasal sides. For the lower vitreous cavity area of WOCT, like the upper one, we selected a B-scan around the area connecting inferotemporal and inferonasal sides with a horizontal line. If the temporal and nasal vortex venous ampullaes could not be connected horizontally, a horizontal line passing through the nasal vortex vein ampulla was used.

・ Line 131-133. What is the area for counting cells in vitreous cavity in a WOCT image? (for example, 13mm x 5mm)

Response

Thank you for pointing out. We added the following.

L150-152

WOCT images were taken horizontally at a maximum of 23 mm and at a maximum depth of 5.3 mm; the average measured vitreous cavity area in 37 cases was 22.5 x 3.0 mm through the macular, 16.8 x 4.4 mm around upper around equatorial area, 15.9 x 3.8 mm around lower.

・ Line 140-142. If you defined “posterior vitreous area” as described here, the area for counting the number of cells varies greatly depending on the height of the PVD, and this is not fair for evaluation of the severity of inflammation in the vitreous cavity.

Response

Thank you for pointing out. We agree your opinion. 

In this study, we investigated patients with or without posterior vitreous detachment. This will be a subject for future consideration.

We added following to “Materials and Methods” and “Discussion” about your opinion.

L163-164

In this study, the presence or absence of posterior vitreous detachment was not considered.

L427-433

Another limitation was various issues with images. The first problem is that the posterior vitreous cavity was selected for image analysis. Strictly speaking, the posterior vitreous cavity area includes the posterior vitreous precortical pocket, which is considered distinct from the vitreous area. We think that it is an important issue that the “posterior vitreous area” that is measured is not fair depending on the presence or absence of posterior vitreous detachment. A distinction between vitreous pockets and posterior vitreous areas should be considered in the future.

・ Line 150-151. The methods for “counting by software” should be described. References for the methods for “counting by software” are necessary.

Response

Thank you for your advices. We added the following.

L176-179

Binarization is a type of image processing that divides the intensity of an image into binary values, such as 0 or 1 [26]. On the WOCT images, cells were binarised and automatically counted by the ImageJ software [27]. The image area to be binarized needs to be manually selected.

26. Nagasato D, Mitamura Y, Egawa M, Kameoka M, Nagasawa T, Tabuchi H, et al. Changes of choroidal structure and circulation after water drinking test in normal eyes. Graefes Arch Clin Exp Ophthalmol. 2019;257(11):2391-9. Epub 20190805. doi: 10.1007/s00417-019-04427-7. PubMed PMID: 31378831.

27. Grishagin IV. Automatic cell counting with ImageJ. Anal Biochem. 2015;473:63-5. Epub 20141224. doi: 10.1016/j.ab.2014.12.007. PubMed PMID: 25542972.

・ Line 167-168. In flow cytometry analyses, it is common to separate cell fractions (erythrocytes, lymphocytes, monocytes, and granulocytes) using forward scatter and side scatter at first. Which cell fraction did you gate on? 

Response

Thank you for your advice. We added the following.

L197-202

They separate cell fractions using forward scatter (FSC) and side scatter (SSC). High score of FSC means that cells in the samples is large. Large cells were identified as gate(A). 

We analyzed using the CD45-SSC gating method. The CD45-SCC gating method uses the fact that immature, proliferative haematopoietic cells express low levels the CD45 antigen. This allows selective analysis of tumor cells by excluding mature normal cells [28].

CD45 gating were enable to analysis CD45 diminish area as Gate1.

・ Line 194-202. Numbers of cells in Group 1 (manual counting) and Group 2 (software-based counting) were quite different and more numbers of cells was counted by manual counting. Why is there such a big difference between manual counting and software-based counting?

Response

Thank you for pointing out. There were difference among Group 1 and Group 2.

Group 1 (manual counting) had the number cells on WOCT less than of Group2.

We've added a discussion about this below to the “Discussion”.

L405-410

 In this study, Group 1 (manual counting) had less the number cells on WOCT than Group 2 (software counting). Also, there were large differences in the number of cells counted between Group 1 and 2. This suggests that software may be able to capture brightness that humans cannot see visually. In all WOCT images of macular, upper and lower, there was a positive correlation between the number of cells counted by humans and the number of cells analyzed by the software.

・ Line 268. “Correlation with IL-6 value” should be “Correlation with IL-6 levels and changing of visual acuity”.

Response

Thank you for your advice. We corrected words.

・ Line 277. “Fig 4. We compared visual acuity between preoperatively and 3 months postoperatively“ should be “Fig 4. Comparisons of IL-6 levels among three groups depending on improvement or deterioration of visual acuity”.

Response

Thank you for your advice. We corrected words.

・ Line 283-291. The finding that there is a correlation between the number of cells in WOCT and CD56-positive cells in FACS is clearly due to the fact that one case of ARN had an exceptionally high number of CD56-positive cells. ARN is infectious uveitis, and the other eight cases are non-infectious. If you want to claim that there is a correlation between the cell count in WOCT and CD56-positive cells in FACS, you should exclude one case of ARN from the analysis.

Response

Thank you for pointing out. We agree your opinion.

We excluded the case of ARN.

Upon reviewing the cases, we found that Case 2 did not have idiopathic uveitis but PIOL.

we fixed it. PIOL is one of non-infectious uveitis, we thought that it was no problem if PIOL included in FACS.

The following was deleted.

L361-362

Only high CD56 score was associated with greater cell counts on WOCT using Pearson's correlation coefficient.

Also, We added the following.

L351-352

There was no statistically significant difference between the observed the number of cells on WOCT and each lymphocyte.

・ Line 311-313. IL-6 may be mainly produced by T cells, B cells and monocytes/macrophages in the eyes with uveitis.

Response

Thank you for your advice. We added following sentences.

L374

IL-6 mainly produced by T cells, B cells and monocytes/macrophages in the eyes with uveitis.

・ Line 329. “OCT is an important tool in ophthalmology” should be “OCT is an important tool in clinical practice in ophthalmology”.

Response

Thank you for your advice. We corrected words.

・ Line 332. “Papers” should be “Articles”.

Response

Thank you for your advice. We corrected words.

・ Line 337. “We evaluated WOCT findings” should be “In this study, we evaluated WOCT findings”.

Response

Thank you for your advice. We corrected words.

・ Line 337-340. The methods for binarization of WOCT image and the methods for counting vitreous cells should be described in Materials and Methods section.

Response

We agree with the reviewer’s opinion. I added the following sentence to the Materials and Methods section. 

L174-176

ImageJ is a Java-based image-processing program developed by the National Institutes of Health and the Optical Computational Metrology Institute. 

・ Line 368. “WOCTA” should be “WOCT”.

Response

Thank you for your advice. We corrected words.

#Reviewer2

・ Keywords should be ordered alphabetical.

Response

Thank you for your advice. We corrected words.

・ Grammar should be checked.

Response

Thank you for your advice. We confirmed grammar and modified.

・ Abstract structure should be the same as described in the journal.

Response

Thank you for your pointing out. We corrected sentences.

・ The inclusion and exclusion criteria determined for the study should be clearly stated in the materials and methods.

Response

Thank you for pointing out. We added sentences the following to the materials and methods.

L75-80

Study inclusion criteria included [1] the presence of posterior vitreous inflammation, [2] unknown cause of posterior uveitis, [3] Cases in which preoperative WOCT and Wide field Optical Coherence Tomography angiography (WOCTA) of En-face could be obtained. Study exclusion criteria included [1] Patients with active systemic inflammation, [2] Cases couldn’t be observed vitreous cavity due to vitreous opacity in preoperative WOCT, [3] The cases couldn’t take En-face images of WOCTA or Optos (Optos® 200Tx, Optos®, Dunfermline, U.K.).

・ The findings given in the table should not be repeated in the text in the results section.

Response

Thank you for your advice. I have deleted the following text.

L211

The 37 patients comprised 18 men and 19 women, with a mean age of 67.5±15.7 years. Baseline logMAR BCVA was 0.22±0.60. Mean IL-6 level was 79.9±7380.9 pg/mL (Table 1).

L216

Uveitis was infectious in 9 cases and non-infectious in 28 cases.

L262-264

(on macular +0.866; upper cavity +0.713; lower cavity +0.568; total vitreous cavity +0.834; P0.001 each). 

L266-269 

In macular images, a correlation was identified (P=0.04; E).

Upper vitreous around the equator with reference to the vortex vena cava (P=0.96; F). 

Lower vitreous around the equator with reference to the vortex vena cava (n=34) (P=0.96; G) and total (sum of macular, upper and lower) (P=0.27 H) (n=34)

L271-276

A correlation was seen for macular images alone (P=0.03; I).

Upper vitreous around the equator with reference to the vortex vena cava (P=0.777; J). 

Lower vitreous around the equator with reference to the vortex vena cava (n=34) (P=0.91; K) and total (sum of macular, superior and inferior) (P=0238; L) (n=34).

＊P0.05

L281-284

No significant difference was seen between upper (PCC+0.09, P=0.96; Figure 2F) and lower images (PCC-0.008, P=0.96; Figure 2G). The number of cells seen in the images in the macular, upper, and lower was examined by totaling, but there was no significant difference (PCC+0.196, P=0.27; Figure2H).

L286-288

Similar to the Human Count Group, no significant difference was observed in studies other than the macular region (PCC+0.048 P=0.777; Figure2J, PCC-0.020 P=0.91; Figure2K, PCC:+0.208 P=0.238 Figure2L).

・ The units of evaluated parameters should be given clearly. For example, pg/mL for IL-6 value.

Response

Thank you for your advice. We added the units of evaluated parameters clearly.

・ The % of demographic characteristics should also be given in Table 1.

Response

Thank you for your advice. We added the % of demographic characteristics in Tabel1.

・ The spelling features of Tables 1 and 2 are not the same.

Response

Thank you for your advice. We corrected the spelling feature of Table 1 and Table2.

・ All p values should be written in italic.

Response

Thank you for your advice. We corrected words.

・ The resolution of the figures is low and should be corrected.

Response

Thank you for pointing out. We corrected the solution of the figures. Also, we added the following to the discussion.

L435-438

The problem with this case series is the WOCT poor images. Images obtained by WOCT were not corrected. This is because it was thought that image correction would become arbitrary. The poor image is one of the problems, however we couldn’t find good idea to solve it.

・ Citations should be corrected according to the journal instructions.

Response

Thank you for pointing out. We corrected according to the journal instructions.

・ The lot number of kits should be stated.

Response

Thank you for pointing out. We added following sentences.

We refrained from adding the kit lot number because it varies depending on the date the sample was collected.

L97

kits for human IL-6 (LUMIPULSE G1200, FUJIREBIO, Tokyo, Japan)

・ The brand and model numbers of the devices used in the study should be stated.

Response

Thank you for pointing out. We added following sentences.

L166

National Institutes of Health, Bethesda, MD, USA

・ What does the word “patiene” in Table 1 mean?

Response

Thank you for pointing out. We corrected the word “patients”.

Participant

Baseline

Characteristics total infectious Non-infectious

Eye,n (%) 37 9 (24.3) 28 (75.7)

Patients,n (%) 34 9 (26.5) 25 (73.5)

Males,n (%) 18 8 (44.4) 10 (55.6)

Females,n (%) 19 1 (5) 18 (95)

Age,years (Mean±SD) 67.5±15.7 67±18.8 67.5±14.2

Baseline logMAR BCVA (Mean±SD) 0.22±0.6 0.15±0.8 0.35±0.5

IL-6 levels, pg/mL (Mean±SD) 79.9±7380.9 1040±12990.6 73.5±990.5

・ Significance levels of p values should be stated below the tables. 

Response

Thank you for pointing out. We added significance levels of p value below the tables.

---

## [Decision Letter · Decision Letter 1]

28 Dec 2023

PONE-D-23-27793R1Relationship between vitreous interleukin-6 levels and vitreous particles findings on widefield optical coherence tomography in posterior uveitisPLOS ONE

Dear Dr. Tagami,

Thank you for submitting your manuscript to PLOS ONE. After careful consideration, we feel that it has merit but does not fully meet PLOS ONE’s publication criteria as it currently stands. Therefore, we invite you to submit a revised version of the manuscript that addresses the points raised during the review process.

We look forward to receiving your revised manuscript.

Kind regards,

Jiro Kogo

Academic Editor

PLOS ONE

Journal Requirements:

Reviewers' comments:

Reviewer's Responses to Questions

**Comments to the Author**

1. If the authors have adequately addressed your comments raised in a previous round of review and you feel that this manuscript is now acceptable for publication, you may indicate that here to bypass the “Comments to the Author” section, enter your conflict of interest statement in the “Confidential to Editor” section, and submit your "Accept" recommendation.

Reviewer #1: (No Response)

2. Is the manuscript technically sound, and do the data support the conclusions?

Reviewer #1: Yes

3. Has the statistical analysis been performed appropriately and rigorously? 

Reviewer #1: Yes

4. Have the authors made all data underlying the findings in their manuscript fully available?

Reviewer #1: Yes

5. Is the manuscript presented in an intelligible fashion and written in standard English?

Reviewer #1: Yes

6. Review Comments to the Author

Reviewer #1: The authors demonstrated that relationship between vitreous interleukin-6 levels and vitreous particles findings on widefield optical coherence tomography in posterior uveitis. The manuscript was well revised according to the reviewers’ suggestions. However, a few mistakes were remained to be corrected.

1. Line 204, line 353 and Fig 5. CD3 was examined in line 356 and Fig 5, whereas CD9 was examined in line 204. CD3 may be correct.

2. Line 344. In Table 4, there were no cases with ARN, thus “ARN acute retinal necrosis” should be erased.

7. PLOS authors have the option to publish the peer review history of their article (what does this mean?). If published, this will include your full peer review and any attached files.

Reviewer #1: **Yes: **Toshikatsu Kaburaki

---

## [Author Response · Author response to Decision Letter 1]

28 Dec 2023

#Reviewer1

1. Line 204, line 353 and Fig 5. CD3 was examined in line 356 and Fig 5, whereas CD9 was examined in line 204. CD3 may be correct.

2. Line 344. In Table 4, there were no cases with ARN, thus “ARN acute retinal necrosis” should be erased.

Response 

Thank you very much for your detailed corrections and advice. The above has been corrected.

---

## [Editor Report · Decision Letter 2]

2 Jan 2024

Relationship between vitreous interleukin-6 levels and vitreous particles findings on widefield optical coherence tomography in posterior uveitis

PONE-D-23-27793R2

Dear Dr. Tagami

We’re pleased to inform you that your manuscript has been judged scientifically suitable for publication and will be formally accepted for publication once it meets all outstanding technical requirements.

Kind regards,

Jiro Kogo

Academic Editor

PLOS ONE

---

## [Editor Report · Acceptance letter]

8 Jan 2024

PONE-D-23-27793R2 

PLOS ONE

Dear Dr. Tagami, 

I'm pleased to inform you that your manuscript has been deemed suitable for publication in PLOS ONE. Congratulations! Your manuscript is now being handed over to our production team.

Kind regards, 

on behalf of

Dr. Jiro Kogo 

Academic Editor

PLOS ONE